# Characterisation of Neurospheres-Derived Cells from Human Olfactory Epithelium

**DOI:** 10.3390/cells10071690

**Published:** 2021-07-04

**Authors:** Elena A. Zelenova, Nikolay V. Kondratyev, Tatyana V. Lezheiko, Grigoriy Y. Tsarapkin, Andrey I. Kryukov, Alexander E. Kishinevsky, Anna S. Tovmasyan, Ekaterina D. Momotyuk, Erdem B. Dashinimaev, Vera E. Golimbet

**Affiliations:** 1Mental Health Research Center, 115522 Moscow, Russia; zelenova@gmail.com (E.A.Z.); lezheiko@list.ru (T.V.L.); golimbet@mail.ru (V.E.G.); 2Sverzhevskiy Otorhinolaryngology Healthcare Research Institute, Moscow Department of Healthcare, 117152 Moscow, Russia; tsgrigory@mail.ru (G.Y.T.); nikio@zdrav.mos.ru (A.I.K.); voroej@gmail.com (A.E.K.); 7svetlana@mail.ru (A.S.T.); 3Koltzov Institute of Developmental Biology of Russian Academy of Sciences, 119991 Moscow, Russia; edm95r@rambler.ru (E.D.M.); dashinimaev@gmail.com (E.B.D.); 4Center for Precision Genome Editing and Genetic Technologies for Biomedicine, Pirogov Russian National Research Medical University, 117997 Moscow, Russia

**Keywords:** olfactory epithelium, neuronal progenitors, olfactory ensheathing cells, RNA expres-sion, maneb

## Abstract

A major problem in psychiatric research is a deficit of relevant cell material of neuronal origin, especially in large quantities from living individuals. One of the promising options is cells from the olfactory neuroepithelium, which contains neuronal progenitors that ensure the regeneration of olfactory receptors. These cells are easy to obtain with nasal biopsies and it is possible to grow and cultivate them in vitro. In this work, we used RNAseq expression profiling and immunofluorescence microscopy to characterise neurospheres-derived cells (NDC), that simply and reliably grow from neurospheres (NS) obtained from nasal biopsies. We utilized differential expression analysis to explore the molecular changes that occur during transition from NS to NDC. We found that processes associated with neuronal and vascular cells are downregulated in NDC. A comparison with public transcriptomes revealed a depletion of neuronal and glial components in NDC. We also discovered that NDC have several metabolic features specific to neuronal progenitors treated with the fungicide maneb. Thus, while NDC retain some neuronal/glial identity, additional protocol alterations are needed to use NDC for mass sample collection in psychiatric research.

## 1. Introduction

Compared to most other disease research, the arsenal of tools for psychiatric research is limited by the lack of reliable biomarkers, the small selection of animal models, and the poor availability of brain tissue. In recent years, studies of genetic and epigenetic factors of mental illness have revealed that most of them are associated with impaired embryonic development of the brain [1,2,3,4]. Since it is impossible to obtain foetal brain samples from future mentally ill people, the study of neuronal stem cells in vitro is the next best thing. This makes the use of neuronal cells in vitro a priority object for model research in psychiatry [5,6]. A common procedure for obtaining such cells is to grow neuronal progenitors and neurons from induced pluripotent cells. This model was used to show a decrease in the expression of protocadherins in schizophrenia [7], impaired β-catenin/BRN2 cascade for idiopathic autism [8], lithium-dependent hyperexcitability [9] and altered calcium signalling pathway [10] in bipolar disorder. Such studies are usually limited to small numbers, often less than 10 samples. The problem with this small sample size is that common mental illnesses are mostly hereditary and highly polygenic [11]. In the past, it was hoped that numerous genetic risk factors would converge into a small number of easily identifiable biological predictors of the disease, which in psychiatric genetics are usually called “endophenotypes” as opposed to the largely subjective behavioural symptoms used in the diagnosis of the diseases [12]. However, it seems that small samples are not enough to reveal endophenotypes, especially for diagnostic purposes. For example, some research groups reported the existence of a presynaptic deficit in the study of induced neurons from cells of patients with schizophrenia [13,14,15], while others have not found such an effect [7,16]. Lack of control for genetic factors may be a source of bias in experimental designs based on diagnostic criteria [17]. In addition, cellular protocols for obtaining neuronal cells from induced pluripotent cells are time-consuming, expensive, and notoriously hard to reproduce [18].

Neural stem cells from adult humans are an alternative source of neuronal stem cells. Neuronal stem cells of the adult brain are few in number and are available for research only post mortem. Another source is the stem cells from the peripheral nervous system. Of these, probably the most studied are the stem cells of the olfactory neuroepithelium, which during a person’s life restore the olfactory receptors in a manner similar to the embryonic central nervous system development [19,20,21,22]. Cells of the olfactory neuroepithelium are of practical importance for studying the loss of smell, especially recently, in the context of COVID-19 infection [23,24], regenerative medicine [25,26], and Parkinson’s disease treatment [27]. In psychiatry, studies based on cultured neural progenitor cells derived from olfactory neuroepithelium (also known as “CNONs”) were used to explore the features of gene expression, epigenetic markers, and the 3D-genome associated with schizophrenia [28,29]. The use of such cells makes it possible to collect more than a hundred individual samples. Alternatively, the neuronal cells from olfactory epithelium could be obtained directly from neurospheres, which in turn could be reliably grown from primarily cultures from biopsies. Such cells (“olfactory neurosphere-derived cells”, ONS) were utilised in a large-scale study of gene expression of schizophrenia and Parkinson’s disease [30]. The authors reported that gene expression in ONS were different and enriched by gene ontologies, for example “ephrin receptor signaling” and “axonal guidance signaling” for schizophrenia, “aryl hydrocarbon receptor signaling” and “purine metabolism” for Parkinson’s disease, and “NRF2-mediated oxidative stress response” for both. Of note, there were no such enrichments when the fibroblasts were used as starting material. These achievements make the approach a promising method to study psychiatric and neurological diseases, especially relevant in genetic context where sample size is the key or in epigenetic context where minimal cell handling is required.

Nasal biopsies usually contain cells from two types of epithelial tissues: olfactory and respiratory. According to the sequencing of transcriptomes of single cells (scRNA-seq), the cytological diversity of the epithelium of the nasal sinuses includes at least 25 cell types [31]. Most of the cells in biopsies are not neuronal, so the researcher’s task is to separate these neuronal cells from the resulting biopsies. Among neuronal cells, nasal biopsies include olfactory receptor cells, olfactory ensheathing cells (OEC), and neuronal stem cells (globose basal cells). Olfactory ensheathing cells are analogues of glial cells for neurons of the olfactory epithelium, which are somewhat similar to the radial glia of the embryonic brain [32]. Therefore, they, as well as neuronal stem cells of the olfactory epithelium, are of interest to the study of mental illness.

The existing protocols involve obtaining neuronal or glial stem cells from neurospheres (NS) that are to be grown from nasal biopsy material on a neuronal selective medium. In a standard medium with serum, cells arise from NS by enzymatic dissociation using trypsin to propagate into an abundant monolayer of cells within two to three weeks, which we further refer to as neurospheres-derived cells (NDC). The aim of this work is to characterize these cells and, if possible, understand how the protocol can be modified to yield more cells with neuronal or glial identity.

## 2. Materials and Methods

### 2.1. Materials and Reagents

#### 2.1.1. Cell Biology

DMEM/F12, cat. C470п (PanEco, Moscow, Russia); Fetal Bovine Serum, qualified, One Shot format, Brazil, Gibco, cat. A3160802 (Thermo Fisher Scientific, Waltham, MA, USA); GlutaMAX Supplement, Gibco, cat. 35050061 (Thermo Fisher Scientific, Waltham, MA, USA); ITS, cat. Φ065 (PanEco, Moscow, Russia); EGF cat. CB-1101001 (PanEco, Moscow, Russia); FGF-2, cat. CB-1102021 (PanEco, Moscow, Russia); Dispase II, Sigma-Aldrich, cat. D4693 (Merck Life Science, Amsterdam, The Netherlands); Antibiotic-Antimycotic (100X), Gibco, cat. 15240062 (Thermo Fisher Scientific, Waltham, MA, USA).

#### 2.1.2. Immunostaining

Purified anti-Nestin Antibody, cat. 656802 (Biolegend, San Diego, CA, USA); Anti-GFAP Antibody, cat. 840001 (Biolegend, San Diego, CA, USA); Anti-TAGLN antibody, Sigma-Aldrich, cat. HPA019467 (Merck Life Science, Amsterdam, the Netherlands); Anti-Actin, α-Smooth Muscle antibody, Mouse monoclonal, Sigma-Aldrich, cat. A5228 (Merck Life Science, Amsterdam, the Netherlands); Goat anti-Rabbit IgG (H+L) Cross-Adsorbed Secondary Antibody, Texas Red, Invitrogen, cat. T-2767 (Thermo Fisher Scientific, Waltham, MA, USA); Alexa Fluor^®^ 488 Goat anti-mouse IgG (minimal x-reactivity) Antibody, cat. 405319 (Biolegend, San Diego, CA, USA).

#### 2.1.3. RNA-seq

RNeasy column kits, cat. 74104 (Qiagen, Hilden, Germany); RNAlater RNA Stabilization Reagent, cat. 76104 (Qiagen, Hilden, Germany); DNase I, cat. M0303 (NEB, Ispwich, MA, USA); CleanRNA Standard kit, cat. BC033 (Evrogen, Moscow, Russia); NEBNext Ultra II Directional RNA Library Prep, cat. E7765 (NEB, Ispwich, MA, USA); NEBNext Poly(A) mRNA Magnetic Isolation Module, cat. E7490 (NEB, Ispwich, MA, USA); NEBNext Multiplex Oligos for Illumina (Dual Index Primers Set 1), cat. E7600 (NEB, Ispwich, MA, USA).

### 2.2. Sample Collection

The study included 11 inpatients (5 women, 6 men, 25–41 y.o.) admitted to the Sverzhevsky Institute to fix the curvature of the nasal septum or hypertrophic rhinitis. Sampling of biological material from the nasal cavity was performed under general anaesthesia in the area of the upper third of the nasal septum (superior turbinate), which is a convenient site for a biopsy that contains most of the olfactory epithelium according to our and other studies [33,34]. Inclusion criteria for the study were age (20–45 y.o.) and the absence of severe somatic and mental illnesses. Exclusion criteria were the absence of chronic rhinosinusitis or acute allergic rhinitis. The collected material was placed in a tube containing 1 mL of DMEM/HAM F12 medium supplemented with foetal bovine serum (10%) and Antibiotic-Antimycotic, Gibco (1%). Samples were delivered to the laboratory within 3 h of biopsy collection.

### 2.3. NS and NDC

NS were obtained according to the protocol described earlier [19,20,21,22]. Biopsies were incubated with Dispase II (2.4 IU/mL) for 1 h at 37 °C. Further, the olfactory epithelium was separated from the lamina propria. Then the lamina propria was divided into samples with a thickness of 200 to 500 µm and each sample was placed in a standard 6-well plate under a cover glass, where it was cultured for 18 days (5% CO_2_, 37 °C) in DMEM/HAM F12 supplemented with 10% foetal calf serum and antibiotics. The medium was changed every 2–3 days. After that, the cells were removed with trypsin and transplanted into flasks coated with poly-L-lysine with a density of 16,000 cells per cm2 to form NS (culture medium based on DMEM/F12 with the addition of GlutaMAX, 1% ITS, 50 ng/mL EGF, 25 ng/mL FGF2). After 15–20 days, the formed neurospheres were collected. Some of them were frozen for RNA isolation using the protective reagent RNALater, the remaining ones were dissociated with trypsin and transplanted onto Petri dishes, cultured in DMEM/F12 medium supplemented with 10% FBS, GlutaMAX, and an antibiotic. After three passages, the cells were harvested and frozen for RNA isolation.

### 2.4. Fluorescent Microscopy

NS obtained according to the protocol described above were collected and dissociated with 0.25% Trypsin-EDTA solution, after which they were planted in a 24-well culture plate treated with poly-L-lysine, using a nutrient medium based on DMEM/F12 supplemented with glutamine, ITS 1%, EGF (50 ng/mL), FGF2 (25 ng/mL), and incubated for a day with 5% CO2, 37 °C. NDCs were seeded in 24-well plates using the standard design described above. NS and NDC were washed three times with PBS and fixed in 4% paraformaldehyde in PBS (pH 7.4) for 15 min at room temperature, after which they were washed again with PBS. Further, the cells were supplemented with primary antibodies diluted (*v*/*v* 1:100) in a “block solution” (DPBS, 10% FBS, 0.1% Triton X-100, 0.01% Tween 20) and incubated for 12 h at 4 °C. After incubation, the cells were washed three times with PBS solution, after which they were supplemented with secondary antibodies diluted (*v*/*v* 1:1000) in a block solution and incubated for 1 h at 37 °C. Then, the cells were washed three times with PBS solution and DAPI solution was added for nuclear staining for 15 min at room temperature. Afterwards, the cells were once again washed with PBS solution. Control wells were processed in the same way, except for the addition of primary antibodies, as well as any antibodies for “pure” control.

### 2.5. RNA-seq and Expression Analysis

RNA was isolated from the cells using RNeasy column kits (Qiagen, Hilden, Germany), then treated with DNase I and purified using the CleanRNA Standard kit (Evrogen, Moscow, Russia). The quality and quantity of the RNA were assessed with the Qubit fluorometer (Thermo Fisher Scientific, Waltham, MA, USA). cDNA libraries were constructed with 125 ng (NS) or 500 ng (NDC) of total RNA using NEBNext Ultra II Directional RNA Library Prep (NEB, Ispwich, MA, USA) with NEBNext Poly(A) mRNA Magnetic Isolation Module and barcoded with dual index primers for pooled sequencing. The sequencing was performed to yield 10–20 million reads (2 × 150 base pairs) per library. The gene-level estimated counts were obtained with the “salmon” tool [35]. The analysis for differentially expressed genes between 11 pairs of NS and NDC was performed with the “DESeq2” R package with sample ID used as a covariate (Love, Huber, and Anders 2014). Transcriptomic cell typing was performed using t-SNE with publicly available datasets; the SRA accession numbers are in Appendix A. Public data were reanalysed from raw FASTQ reads with salmon the same way as described above. For plots on Figures 2B and 4 we used combined TPM (transcripts per million) normalised data filtered by ensembl genes which have TPM > 5 in at least 90% of all datasets. The log-transformed TPM values were used for dimensional reduction. We utilised R packages “Rtsne” for t-SNE with default settings (Barnes–Hut implementation of t-SNE with initial PCA with 50 principal components and perplexity parameter specified in the text) and “umap” for UMAP with default settings (“naive” method). The same data were used for Spearman’s rank correlation computations. Gene Set Enrichment Analysis was performed with “msigdbr” and “clusterProfiler” R packages with default parameters on a list of genes, ordered by -log(p-level,adjusted) * sign(FC), where *p*-values and FC (fold change of expression) were taken from Deseq2 results. Net plots visualisations were generated with the “enrichplot” R package [36] and the visualisation of biochemical pathways in Appendix A was created with the help of “pathview” R package [37].

## 3. Results

We collected nasal biopsies from eleven people. We then obtained NS from the collected material and utilised a protocol described in the Methods section to grow a monolayer of NDC. NDC on serum-rich media produce an abundant monolayer of cells, which can be harvested and stored in nitrogen for later use.

NDC and NS were both stained with vascular markers, SMA and TAGLN (Figure 1), but NDC cultures have a portion of cells with a much more prominent SMA signal (Figure 1A and Appendix A). The expression data support this observation: NDC have much higher *ACTA2*, SMA gene, (log2FC = 3.65, *p* = 4.46 × 10^−22^), and *TAGLN* expression (log2FC = 4.87, *p* = 9.5 × 10^−43^). It seems that while NDC still retain glial (GFAP) and neuronal (beta-tubulin and nestin) markers, they are shifted more towards other cell types.

From the expression analysis, it is clear that the transcriptomes of NS and NDC are different (Figure 2A). To ascertain the cellular identity of NDCs, we compared the obtained transcriptomes with previously published bulk RNAseq data of different neural progenitor cells (NPC), glial, nasal, and other (listed in Appendix A, note that there are no available RNAseq data for OEC). The clustering picture of transcriptomes on t-SNE plot suggests that while NS have transcriptomes very similar to previously published expression data from nasal biopsies, NDC are closer to the transcriptomes of cells from smooth muscle tissue and fibroblasts and not NPC (Figure 2B). Rank correlations between tested expression profiles reveal that NS to NDC transition leads to loss of similarity with glial cells (ρ falls from 0.74 to 0.58 on average) and NPC (0.51 to 0.38). NDC, like NS, remain similar with smooth muscle cells (0.87 to 0.78 with maximum similarity with fetal aorta cells, 0.82 to 0.84 [38]) and to less extend with fibroblasts (0.82 to 0.73) (Appendix A). Note that the same analysis for Matigian et al. array expression data suggests that expression profiles from ONS also resemble data from fibroblasts more than NPC, glial cells, and cells in this study (Appendix A).

The differential expression analysis (Deseq2) of NDC vs. NS revealed that there were multiple changes in gene expression (Figure 2C, Appendix A). We had enough data for gene set enrichment analysis (GSEA) which we used to elaborate the processes involved in NS to NDC transformation. GSEA on gene ontology collection of cellular components (GO:CC) shows that energy and protein metabolism, followed by ontologies related to neuronal components, are affected the most during NS to NDC transition (Appendix A). GSEA on gene ontology collection of biological processes (GO:BP) revealed a plethora of affected pathways (Appendix A). There are three main themes among them: upregulated in NDC metabolic processes (mainly energetic), downregulated vascular transformation, and neuronal morphogenesis (Appendix A). The same three themes are present in the GSEA analysis of pathway enrichment for KEGG [39] (Appendix A), WikiPathways [40] (Appendix A), and MSigDB Hallmark [41] (Appendix A) pathways collections. Some representative GSEA enrichment score curves are presented in Figure 3A. The interaction of affected genes in the “vascular” and “neuronal” categories are presented on the net plot in Figure 3B. Note that both groups of genes contain considerable intersection, involved both in neuro- and vasculogenesis, namely *MAPK1*, *VEGFA*, *APOE*, *NRP1,* and others.

To understand why NDC are further from neuronal cell identity than NS, we compared our cells with a large publicly available cytotoxicity RNAseq study of in vitro 2D-cultures of neuronal progenitors [42]. Various dimensional reduction plots demonstrate that the transcriptomic position of NPC treated with the fungicide maneb (manganese ethylene-bis-dithiocarbamate) is very robustly found colocalized with positions of data from NDC (Figure 4). Another notable feature (less prominent by a wide margin though) is cadmium-treated NPC; the effect is best seen on PCA plot (Figure 4B). Spearman’s rank correlation of expression profiles of NDC and Kuusisto et al. data were not the highest for NPC treated with maneb (the highest correlation was for trans-retinoic acid, ρ = 0.73), but for maneb we observed the highest and the only major increase in similarity from NS to NDC (0.59 to 0.7) for all of the tested datasets.

## 4. Discussion

Our results show that the key problem of growing cells of neuronal origin from NS obtained from the olfactory epithelium is the balance between neural cells and non-neural cells (vascular or stromal cells, for example). The cells, originating from NS, often end up as a mixture of stem cells [43]. It is possible to further optimise cell development towards desired types. For example, in an article by Begum et al., the production of neurons from neurospheres was achieved by increasing the concentration of CO_2_ with a serum-free medium at the stage of neurospheres [44]. Possibly, under conditions of greater hypoxia with less reactive oxygen species (ROS) production, there are fewer stimuli for cells in the medium to develop towards vascularisation and angiogenesis [45,46,47].

Neuronal/glial and vascular markers are not mutually exclusive. For example, NPC express the smooth muscle marker SMA in mice which control their migration [48]. It was shown that OEC also express SMA and that the proportion of OEC simultaneously expressing SMA and GFAP grows with time of cultivation on the cell culture of model animals: rats, mice, hamsters, and monkeys [49,50]. Similar cells with glial, vascular, and partially neuronal markers are present with the production of spheres from other sensory organs in animals: mice hair follicle [51], porcine retina [52], and mice tongue fungiform papilla [53]. The multipotent state of those cells seems to be essential for sensory organ regeneration. The property is exploited in regenerative medicine. These kinds of cells were successfully employed for spinal injury therapy in animal models [54,55] and even in some human cases [25,26,56]. The particular combination of glial (GFAP) and vascular (SMA) markers was also observed for gliosarcomas, a subtype of gliomas [57,58,59]. It was found that the vascular component of these gliomas bears the same somatic p53 mutations as glial cells, pointing to the monoclonal origin of the glial and vascular components [60,61]. We observed that major cluster of ontologies of downregulated genes in NDC consists of both vascular and neuronal processes, with many shared genes (Figure 3B, Appendix A), suggesting that in the cells of this study, the vascular markers were acquired from or even present in the original NPC and/or OEC in the olfactory biopsies.

The apparent difference between expression in NDC and Matigian et al. ONS is worth consideration. There are small but important discrepancies in protocols that could contribute to a difference in ONS and NDC. Probably the most important of these is that in the Matigian et al. protocol, primary cultures were grown in suspensions from enzymatically and mechanically dissociated biopsies. It is a suitable method for targeting the numerous neuroblasts of the olfactory epithelium of rats and mice, but it may not be very effective in humans. We used the explant method, which is supposed to be more gentle. It is also possible that both protocols work well, but stem cells from NS are unstable in their undifferentiated state, and small changes in their handling can lead to completely different outcomes.

We found that our RNA-seq NDC transcriptomes are strikingly similar with NPC transcriptomes treated with maneb. Maneb with paraquat is the basis of the environmental risk Parkinson’s animal model (“MNPQ model”) [62]. The Parkinson’s pathway is the second of the most enriched KEGG ontologies in our GSEA analysis (Appendix A), but its interpretation is not straightforward. In NDC, several principal Parkinson’s markers are decreased relative to their NS expression, namely α-synuclein gene, *SNCA* (log2FC = –2.38, *p* = 0.003), *PTGDS* (log2FC = –5.48, *p* = 1.7 × 10^−38^), *LRRK2* (log2FC = –3.09, *p* = 1.47 × 10^−18^) [63]. Whatever stalls the neuronal progression in NDC is probably not directly related to Parkinsonian pathological process.

The reason that transcriptomes of NDC resemble NPC treated with maneb could be elevated ROS production (note, that the most enriched KEGG ontology in GSEA was “oxydative phoshorilation”). Indeed, the cytotoxicity of both maneb and cadmium is linked to elevated levels of ROS [64,65,66]. NDC have increased expression levels of genes of glutathione peroxidase, *GPX1* (log2FC = 1.73, *p* = 2.99 × 10^−14^) and superoxide dismutase 1, *SOD1* (log2FC = 0.92, *p* = 1.02 × 10^−10^). As we mentioned earlier, transient ROS production is known to promote vascular growth, which is what we see in the case of NS to NDC progression. The counterpoint is that many other substances in the Kuusisto et al. screen are known to cause oxidative stress, like lead [67], permethrin [68], and isotretinoin [69]. Furthermore, Roede et al. reported that in SH-SY5Y neuroblastoma cell line, it is paraquat rather than maneb that is responsible for ROS production [70]. Anderson et al. studied the biochemistry of acute treatment of neuroblastoma cell line SK-N-AS with maneb [71,72]. They found that while energy metabolism is affected by maneb, the effect is more specific. Maneb seems to act on thiol groups of mitochondrial proteins and alter glucose metabolism by promoting gluconeogenesis at the expense of ATP production. In our NDC, we also observed multiple changes in the expression of genes involved in energetic and biochemical pathways, namely oxidative phosphorylation, nucleotide metabolism, and the citrate cycle (Appendix A). Still, the origin of the similarity of expression profiles in NDC and NPC treated with maneb is unclear and worthy of additional investigation.

There are some limitations to this study. First, the presented RNAseq results are acquired on bulk RNA and are not aware of the heterogeneity of NDC and NS. The immunofluorescent pictures do confirm that most of the cells bear the same markers but to a different extent, like in the case of SMA. Different methods, such as scRNA-seq, are needed to study the contribution of different types of cells in the observed effects. Second, we used dimensionality reduction of transcriptome data to contrast obtained expression profiles against publicly available data. While data from similar cell types tend to cluster together, there is no way to correct for differences in technicalities used to produce data in different laboratories and interpretation of such results should be conducted with caution. On the other hand, our second method of expression profiles comparison with rank correlations could be too conservative and misleading when correlations are too small. Most importantly, the results concerning the similarity of NDC with NPC treated with maneb are based on a single observation, which should be independently replicated to draw a definitive conclusion. We plan to address this in future research.

In conclusion, we observed that gene expression profiles of NDC with present cultivation routine tend to alter their energetic metabolism, which reflects increased vascular and stromal processes associated with a partial loss of their neuronal identity. For now, it seems more practical to use original NS for studying ex vivo neuronal cells. Still, it is possible that by growing NDC in the presence of ROS, scavengers and/or different sources of energy cell cultures with more pronounced neuronal signatures can be obtained to render the material more useful in mass collection for psychiatric research.

## Figures and Tables

**Figure 1 cells-10-01690-f001:**
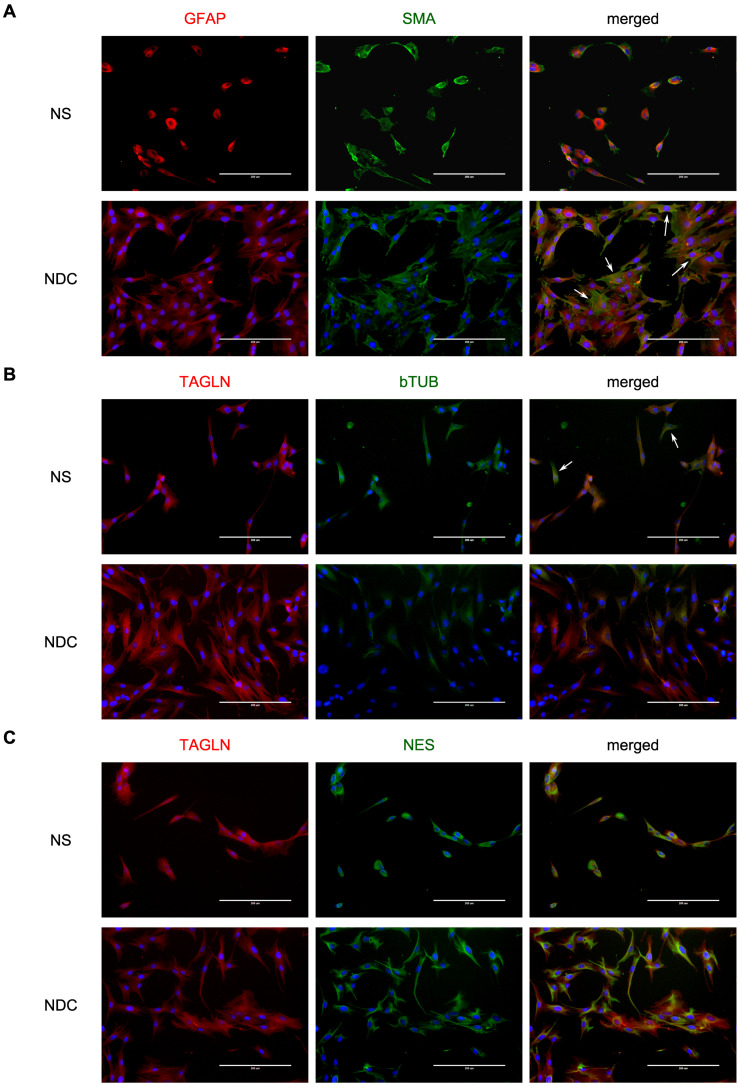
Vascular, glial, and neuronal markers of neurospheres (NS) and neurospheres-derived cells (NDC). (**A**) NS and NDC are stained with GFAP and smooth-muscle actin alpha (α-SMA). The arrows indicate NDC with more pronounced SMA staining. (**B**) NS and NDC are stained with transgelin (TAGLN) and beta-III-tubulin. The arrows indicate cells from NS with beta-III-tubulin staining. (**C**) NS and NDC are stained with transgelin (TAGLN) and nestin (NES). On all panels where the blue colour is present, it corresponds to DAPI dye.

**Figure 2 cells-10-01690-f002:**
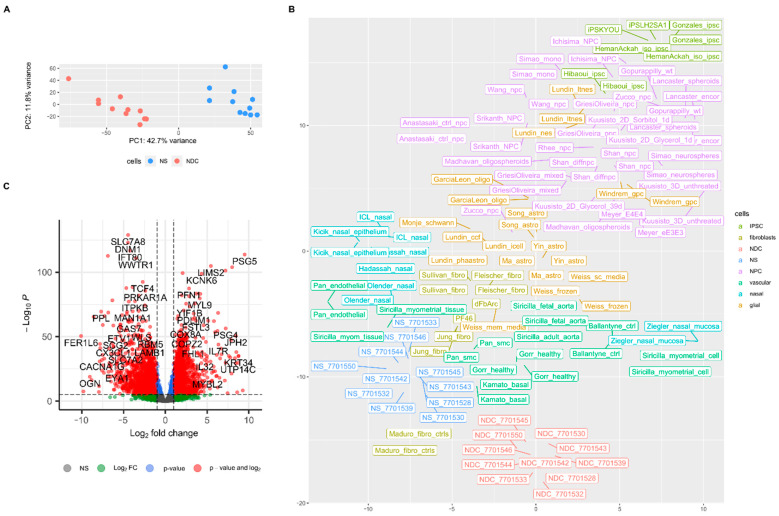
NDC and NS transcriptomes. (**A**) Principal component analysis of NDC and NS transcriptomes. (**B**) t-SNE plot (with perplexity parameter, *p* = 15) of NS, NDC, and other transcriptomes from: NPC and mixed stem neuronal cells, IPSC, fibroblasts, various cells from nasal biopsies, glial cells, vascular cells, and fibroblasts; the complete list is in Appendix A. (**C**) Volcano plot of NDC vs. NS results of differential expression analysis with Deseq2.

**Figure 3 cells-10-01690-f003:**
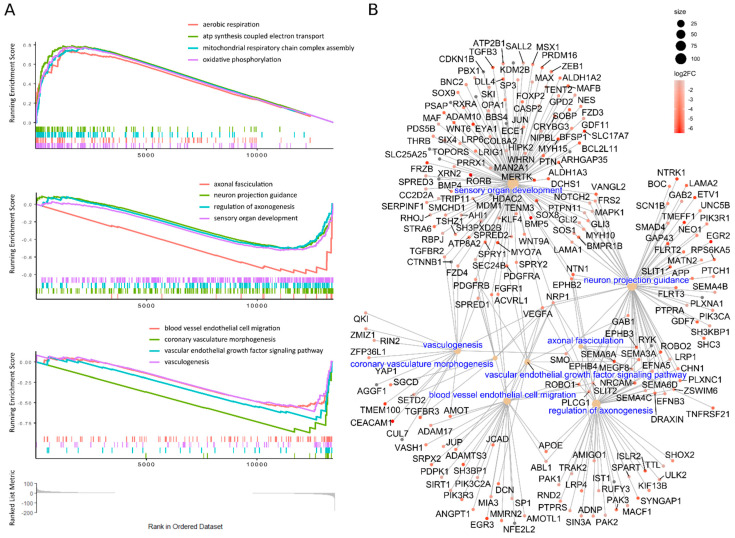
GSEA of GO: BP results of NDC vs. NS expression data. (**A**) Representative GSEA curves of significant (*p* < 0.05, Benjamini–Hochberg adjusted) biological processes: energy metabolism, neuronal, and vascular (top to bottom). Coloured stroke panels represent occurrences of individual genes, belonging to corresponding gene sets. Running enrichment scores are built on genes, ordered by ranking score [-log (Deseq2 p-level, adjusted) * sign (FC)]; the common scale is depicted at the bottom of the picture. The genes on the left are expressed more in NDC while those on the right in NS. (**B**) Network of downregulated in NDC individual genes, which belong to various neuronal and vascular GO:BP ontologies, enriched in GSEA (ontologies’ names are in blue). The colour corresponds to log2(FC); the size of the node is proportional to the size of the gene set.

**Figure 4 cells-10-01690-f004:**
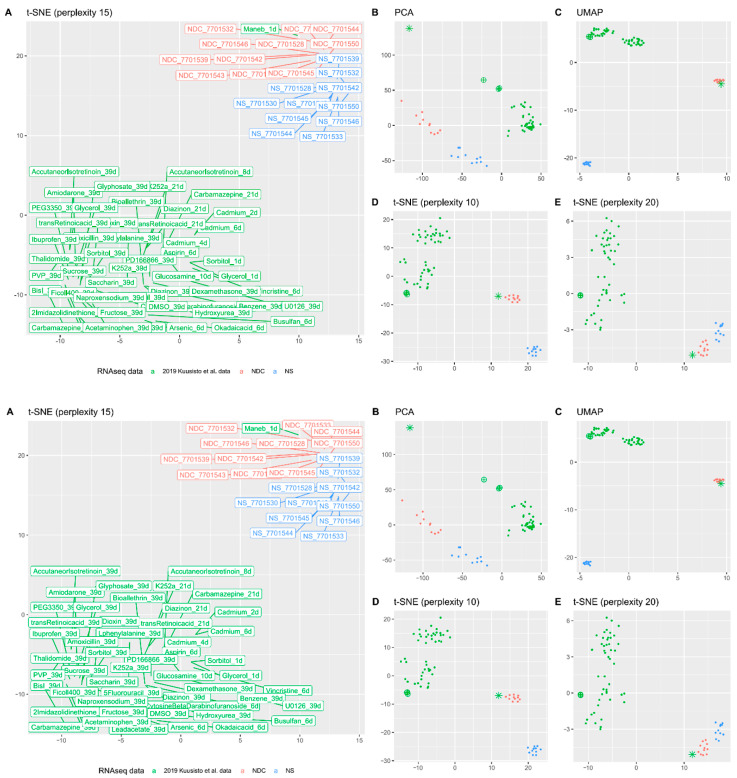
Transcriptomic typing with Kuusisto et al. datasets [42]. (**A**) The t-SNE plot (with perplexity parameter, *p* = 15) of transcriptomic data of NS and NDC with data from the Kuusisto et al. dataset. A subset of the data was used for this picture—all substances with the longest time of exposure (up to 39 days). For cadmium, there are three points (50 µM exposure for two, four, and six days) and the only available dataset for maneb is exposure with 60 µM of the substance for one day. (**B**–**E**) The same picture is visualised with PCA (**B**), UMAP (**C**), and t-SNE with *p* = 10 (**D**) and *p* = 20 (**E**). The point corresponding to maneb RNA-seq data is represented as an asterisk and the cadmium data as crossed circles.

## Data Availability

Raw RNA-seq sequencing reads generated in this study were deposited to the SRA database, accession number PRJNA732358 (https://www.ncbi.nlm.nih.gov/sra/PRJNA732358, accessed on 3 July 2021).

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
