# Peer review of "Characterisation of Neurospheres-Derived Cells from Human Olfactory Epithelium"

_cells, 2021, doi:10.3390/cells10071690_

Round 1

Reviewer 1 Report

Neurospheres obtained from nasal biopsies and neurosphere-derived cells are potentially important and certainly under-appreciated models to study neurodevelopmental aspects of disorders, particularly psychiatric disorders. Thus characterization of transcriptome profiles of these cells is a welcomed addition for the field. While the authors mostly focused on potential use of these cells for studying psychiatric disorders, autologous cells with abilities to differentiate into neuronal and probably other kinds of cells may have applications in regenerative medicine. The study was performed using mostly adequate methods with reasonable data analyses. However, there are several points in the manuscripts which have not clearly articulated, and some methods and conclusions need either better explanation or should be corrected.

The topic is not novel, and there is similar study (doi: 10.1242/dmm.005447) published in 2010, which was performed on larger number of samples albeit using inferior microarray gene expression profiling. Although the authors cited studies of this team, but unfortunately not this particular study. Comparison of results would be very helpful, if not crucial, for discussion.

Both studies considered utilities of these cellular models for studying brain disorders. However, while the authors of the article of 2010 concluded that “They will be useful for understanding disease aetiology, for diagnostics and for drug discovery”, conclusion in the manuscript is not clear: “It is possible that by growing NDCs in the presence of ROS scavengers and/or different sources of energy cell cultures with more pronounced neuronal signatures can be obtained to render the material useful in mass collection for psychiatric research.” It does not sound like substantial contribution to the field, and it is certainly a step back as compared to the conclusion of their predecessors. I disagree with the author’s conclusion, and I think it is based in the wrong assumption that iPSC-derived neural progenitors are the perfect model to study brain disorders.

Statement that neurospheres (cells of presumably neural lineage) obtained from nasal biopsies in culture differentiate into vascular cells (epithelial or smooth muscle) is non-trivial and requires stronger support. It seems that this conclusion was based on marker and GSEA analyses. In my opinion the classification of cell types using markers is more complicated than what is being presented by the authors. Although there are widely accepted markers of different cell types, using highly multiplexing approaches in protein and RNA detection as well as transcriptome-wide expression profiling demonstrated that only a few of them are highly specific, while others should be used only in the right context. This is more important when describing unknown or under-investigated cell types, and especially in vitro, where expression of some genes may be altered by environment radically different from conditions in vivo.

For example, ACTA2 (the gene encoding alpha-SMA) is expressed in multiple tissues and cell types (see for example Human Cell Protein atlas). Although it is highly expressed in vascular smooth muscle cell types, is is also true for other smooth muscles and it is considered as a marker of myofibroblasts. More importantly, it is also shown to be involved in migration of neural stem cells (https://doi.org/10.1155/2020%2F4764012), and it could conceivably have higher expression in neural progenitors, cells with more pronounced migration properties. It is suffice to say that the expression of gene with such a widespread expression pattern in neurosphere-derived cells is not a strong indication that cells are becoming more “vascular”.

Similarly, enrichment analysis showing several GO terms does not necessarily demonstrate differentiation into vascular type of cells but could instead indicate similarity in differentiation processes of neuronal and vascular cell types.

Thus, the authors did not provide convincing evidence that the neurosphere-derived cells are differentiating toward vascular type (this is one one of the author’s conclusion).

The authors mentioned using covariates for age and gender. If I understand study design correctly, both groups contain the same number of samples developed from the same individuals. If this is correct, age and gender corrections do not seem necessary; moreover, pairwise (more powerful) analysis is required. Otherwise, the authors need to clarify experimental design. If both groups contain independent samples, than 1) it is necessary to describe age and sex of recruited individuals, 2) define if age and gender contribute significantly to variation necessitating correction, and 3) test for existence of other important confounding factors and make correction if necessary. Surrogate variable analysis is a popular approach to analyze and correct for confounding factors, but certainly there are other approaches.

tSNE analysis of author’s data combined with data from another study reauires more explanation. I assume that tSNE was based on principal components (how many?). Did the authors unite the data and if so at what level – fastq files, BAM files, counts, normalized reads, etc. to proceed to perform PCA, or principal components from referenced study were used to project the author’s own data? I think many people will argue that comparison of two different datasets using the first or even either approach is incorrect (as the difference in sample handling and library preparation will create a bias due to batch effect), so justification is needed. I do not think many readers (including me) have deep understanding how tSNE works, but there is a common believe that “distances between clusters might not mean anything” (https://distill.pub/2016/misread-tsne/).  I found the results somewhat compelling, but without sufficient method details and justification they are not convincing enough.

In general, it is not clear what was the motivation to compare expression profiles of neurosphere-derived cells with neurotoxicity screening data performed on neural progenitor cells. Although the authors did their best to discuss the results, it does not seem that alteration of gene expression of iPSC-derived neural progenitors by toxic substances is a straightforward approach to describe the nature of NDC.

Overall, the data presented by the authors is an important contribution to the field, but discussion and conclusions drawn from the data seems superficial and could be explored further.

Author Response

R1: Neurospheres obtained from nasal biopsies and neurosphere-derived cells are potentially important and certainly under-appreciated models to study neurodevelopmental aspects of disorders, particularly psychiatric disorders. Thus characterization of transcriptome profiles of these cells is a welcomed addition for the field. While the authors mostly focused on potential use of these cells for studying psychiatric disorders, autologous cells with abilities to differentiate into neuronal and probably other kinds of cells may have applications in regenerative medicine. The study was performed using mostly adequate methods with reasonable data analyses. However, there are several points in the manuscripts which have not clearly articulated, and some methods and conclusions need either better explanation or should be corrected.

The topic is not novel, and there is similar study (doi: 10.1242/dmm.005447) published in 2010, which was performed on larger number of samples albeit using inferior microarray gene expression profiling. Although the authors cited studies of this team, but unfortunately not this particular study. Comparison of results would be very helpful, if not crucial, for discussion.

A: We completely agree, this is a very relevant paper. We added the consideration of this work to the Introduction and Discussion sections. We also compare the data from this paper with our data, results are presented on a new Supplementary Figure 2B.

The part in the Introduction:

“Alternatively, the neuronal cells from olfactory epithelium could be obtained directly from neurospheres which in turn could be reliably grown from primarily cultures from biopsies. Such cells ("olfactory neurosphere-derived cells", ONS) were utilised in a large-scale study of gene expression of schizophrenia and Parkinson disease (Matigian et al. 2010). The authors reported that gene expression in ONS were different and enriched by gene ontologies, for example "ephrin receptor signaling" and "axonal guidance signaling" for schizophrenia, "aryl hydrocarbon receptor signaling" and "purine metabolism" for Parkinson disease and NRF2-mediated oxidative stress response for both. Of note there were no such enrichments when the fibroblasts were used as starting material. These achievements make the approach a promising method to study psychiatric and neurological diseases, especially relevant in genetic context where sample size is the key or in epigenetic context where minimal cell handling is required.”

The part in the Discussion:

“The apparent difference between expression in NDC and Matigian et al. ONS is worth consideration. There are small but important differences in protocols that could contribute to a difference in ONS and NDC. Probably the most important of these is that in the Matigian et al. protocol, primary cultures were grown in suspensions from enzymatically and mechanically dissociated biopsies. It is a suitable method for targeting the numerous neuroblasts of the olfactory epithelium of rats and mice, but it may not be very effective in humans. We used the explant method, which should be more gentle. It is also possible that both protocols work well, but stem cells from NS are unstable in their undifferentiated state, and small changes in their handling can lead to completely different outcomes.”

R1: Both studies considered utilities of these cellular models for studying brain disorders. However, while the authors of the article of 2010 concluded that “They will be useful for understanding disease aetiology, for diagnostics and for drug discovery”, conclusion in the manuscript is not clear: “It is possible that by growing NDCs in the presence of ROS scavengers and/or different sources of energy cell cultures with more pronounced neuronal signatures can be obtained to render the material useful in mass collection for psychiatric research.” It does not sound like substantial contribution to the field, and it is certainly a step back as compared to the conclusion of their predecessors. I disagree with the author’s conclusion, and I think it is based in the wrong assumption that iPSC-derived neural progenitors are the perfect model to study brain disorders.

A: We indeed are not content with NDC as a psychiatric cell model. We expected to grow cells, which are closer to their neuronal identity, because that was a motivation in the first place. In this regard, original NS clearly are better. We tried our best to use data from this experiment to understand what could be done to push NS to a more desirable path. Maybe other researchers in a similar situation could find this helpful. We agree that iPSC-derived neural progenitors are not always the perfect model for psychiatric research. For example, the epigenetic information is lost during reprogramming, which otherwise could be explored with cells such as NPC or OEC from nasal biopsies. We added this consideration to the Introduction.

Matigian et al. (2010) made a conclusion on the utility of cells cultivated from NS from nasal biopsies while we try to be more cautious. The authors saw an enrichment of relevant gene ontologies by comparing cells from donors with schizophrenia and Parkinson disease with cells from healthy donors and, importantly, that no such enrichment was observed when fibroblasts were starting cell material. We indeed planned to do something similar, but we did not conduct such experiments and cannot draw such strong conclusions. Also of note, while the protocol from Matigain et al. paper is similar to ours, the data seem to be quite different. The Spearman’s rank correlation analysis points that Matigain et al. ONS resemble RNAseq data from fibroblasts more than from our NDC. So the direct comparison of our results is not as straightforward as expected.

We agree with the concern and try to tone down the conclusions in a revised version. In the Abstract the sentence “We found that they are most probably set in a state of transition from glial and neuronal to vascular identity” was replaced with “We found that processes, associated with neuronal and vascular cells, are downregulated in NDC. Comparison with public transcriptomes revealed a depletion of neuronal and glial components in NDC.” The conclusion section of the Discussion was rewritten: “We observed that gene expression profiles of NDC with present cultivation routine tend to alter their energetic metabolism, which reflects increased vascular and stromal processes associated with a partial loss of their neuronal identity. For now it seems more practical to use the original NS for studying ex vivo neuronal cells. Still, it is possible that by growing NDCs in the presence of ROS scavengers and/or different sources of energy cell cultures with more pronounced neuronal signatures can be obtained to render the material more useful in mass collection for psychiatric research.”

R1: Statement that neurospheres (cells of presumably neural lineage) obtained from nasal biopsies in culture differentiate into vascular cells (epithelial or smooth muscle) is non-trivial and requires stronger support. It seems that this conclusion was based on marker and GSEA analyses. In my opinion the classification of cell types using markers is more complicated than what is being presented by the authors. Although there are widely accepted markers of different cell types, using highly multiplexing approaches in protein and RNA detection as well as transcriptome-wide expression profiling demonstrated that only a few of them are highly specific, while others should be used only in the right context. This is more important when describing unknown or under-investigated cell types, and especially in vitro, where expression of some genes may be altered by environment radically different from conditions in vivo.

For example, ACTA2 (the gene encoding alpha-SMA) is expressed in multiple tissues and cell types (see for example Human Cell Protein atlas). Although it is highly expressed in vascular smooth muscle cell types, is is also true for other smooth muscles and it is considered as a marker of myofibroblasts. More importantly, it is also shown to be involved in migration of neural stem cells (https://doi.org/10.1155/2020%2F4764012), and it could conceivably have higher expression in neural progenitors, cells with more pronounced migration properties. It is suffice to say that the expression of gene with such a widespread expression pattern in neurosphere-derived cells is not a strong indication that cells are becoming more “vascular”.

A: We tried to employ RNAseq analysis for cell typing because we felt that IF markers in this case do not tell a definitive story. While ACTA2 could be important for NPC, we used another smooth muscle cell marker TAGLN which has a more narrow expression profile. It could be clearly seen from the data of the Human Cell Protein Atlas. The point for vascularisation was not only from marker staining but also the appearance of cells with much more pronounced ACTA2 expression. We added additional pictures in Supplementary files to illustrate this more clearly.

R1: Similarly, enrichment analysis showing several GO terms does not necessarily demonstrate differentiation into vascular type of cells but could instead indicate similarity in differentiation processes of neuronal and vascular cell types.

Thus, the authors did not provide convincing evidence that the neurosphere-derived cells are differentiating toward vascular type (this is one one of the author’s conclusion).

A: We rephrased the text to emphasize that the NDC are somewhat similar to vascular cell type, and not vascular. We added fibroblasts to Figure 2B to underline similarity between NS/NDC with stromal cells and to make presented results consistent with data on Figure S2. We agree that there is indeed a similarity in neuronal and vascular cell processes, we tried to illustrate this with Figure 3B and S4. We added this point to the Discussion.

R1: The authors mentioned using covariates for age and gender. If I understand study design correctly, both groups contain the same number of samples developed from the same individuals. If this is correct, age and gender corrections do not seem necessary; moreover, pairwise (more powerful) analysis is required. Otherwise, the authors need to clarify experimental design. If both groups contain independent samples, than 1) it is necessary to describe age and sex of recruited individuals, 2) define if age and gender contribute significantly to variation necessitating correction, and 3) test for existence of other important confounding factors and make correction if necessary. Surrogate variable analysis is a popular approach to analyze and correct for confounding factors, but certainly there are other approaches.

A: We fully agree with the Reviewer that there is no need to use age and gender as covariates because the presented data are from the cell material from the same individuals. We re-analysed the data using individuals as a covariate. The Methods section was updated: “The analysis for differentially expressed genes between 11 pairs of NS and NDC was performed with the “DESeq2” R package with sample ID used as a covariate”. We also updated the figures (2C, 3, and Supplementary figures), tables (S2-7) and numbers in the text according to the new analysis. The new results are different in a sense that they have more significantly differentially expressed genes, 9093 new vs 8906 original DE genes with adjusted p-value < 0.05, with 95% of them being the same. The conclusions remain mostly unchanged, since the top categories in GSEA remain standing. However we removed “gluconeogenesis” in the Discussion since the corresponding term became less significant with the new analysis. The term was replaced with “nucleotide metabolism” since purine/pyrimidine biochemical pathways enrichment is more prominent in most of our GSEA.

R1: tSNE analysis of author’s data combined with data from another study requires more explanation. I assume that tSNE was based on principal components (how many?). Did the authors unite the data and if so at what level – fastq files, BAM files, counts, normalized reads, etc. to proceed to perform PCA, or principal components from referenced study were used to project the author’s own data? I think many people will argue that comparison of two different datasets using the first or even either approach is incorrect (as the difference in sample handling and library preparation will create a bias due to batch effect), so justification is needed. I do not think many readers (including me) have deep understanding how tSNE works, but there is a common belief that “distances between clusters might not mean anything” (https://distill.pub/2016/misread-tsne/).  I found the results somewhat compelling, but without sufficient method details and justification they are not convincing enough.

A: We add a text in the Methods section which describes data preparation for the t-SNE and UMAP analysis. “Public data were reanalysed from fastq with salmon the same way as as described above. For plots on Figures 2B and Figure 4 we used combined TPM (transcripts per million) normalised data, filtered by ensembl genes which have TPM > 5 in at least 90% of all datasets. The log-transformed TPM values were used for dimensional reduction. We utilised R packages “Rtsne” for t-SNE with default settings (Barnes-Hut implementation of t-SNE with initial PCA with 50 principal components and perplexity parameter specified in the text) and “umap” for UMAP with default settings (“naive” method)”.

We definitely agree that dimensionality reduction on transcriptomic data is far from perfect for cell typing, and, of course, the interpretation of distance between clusters is not straightforward. It is still better than markers though. The data on Figure 2B could speak for themselves. All of the data were collected by different laboratories with different methods, but still every major cell type is clustered together.

To provide an alternative to dimensionality reduction plots we add probably the most conservative way to compare expression profiles via Spearman’s rank correlation which is also used to compare our results with the Matigian et al data. The results are presented in a new Figure S2A and are in general in agreement with t-SNE plot on Figure 2B.

R1: In general, it is not clear what was the motivation to compare expression profiles of neurosphere-derived cells with neurotoxicity screening data performed on neural progenitor cells. Although the authors did their best to discuss the results, it does not seem that alteration of gene expression of iPSC-derived neural progenitors by toxic substances is a straightforward approach to describe the nature of NDC.

A: We alter the text to better reflect our motivation. Now the corresponding results section goes like this:

“To understand why NDC are being further from neuronal cell identity, we compared our cells with a large publicly available cytotoxicity RNAseq study of in vitro 2D-cultures of neuronal progenitors [40]. Various dimensional reduction plots demonstrate that the transcriptomic position of NPC treated with the fungicide maneb (manganese ethylene-bis-dithiocarbamate) is very robustly found colocalized with positions of data from NDC (Figure 4). Another notable feature (less prominent by a wide margin though) is cadmium-treated NPC, the effect is best seen on PCA plot (Figure 4B). Spearman’s rank correlation of expression profiles of NDC and Kuusisto et al. data were not the highest for NPC treated with maneb (the highest correlation was for trans-retinoic acid, ρ=0.73), but for maneb we observed the highest and the only major increase in similarity from NS to NDC for all of the tested datasets (0.59 to 0.7).”

R1: Overall, the data presented by the authors is an important contribution to the field, but discussion and conclusions drawn from the data seems superficial and could be explored further.

A: Thank you for your helpful comments, we tried our best to address them in a revised version.

Reviewer 2 Report

The authors realized nasal biopsies to generate neurospheres (NS) from which they obtain neurospheres-derived cells (NDC). The authors characterised and compare NS and NDC using immunofluroescence and RNA-seq expression profile. The authors found that the NDC are in a transition state of differenciation from a glial/neural to a vascular pre-profile. They also discover that the NDC exert some metabolic features, that they compared to neuronal progenitor treated with a fongicide (maneb). The authors concluded that additional protocols are needed to use NDC for mass sample collection in psychatric diseases.

The research was conducted correctly and the authors present possible further steps that are necessary to accomplish to verify their findings. They also point out the limitations of their study.

I have minor suggestions:

Introduction - The authors should provide a mors detailed description and/or explanation of the cells they considers. Indeed, it is really confusing to cite olfactory ensheating cells (OEC) without mentionnig the olfactory stem cells (OSC). The end of the introduction has to be revised according to this suggestion. The confusion is reinforced later because authors described olfactory stem cells obtention (Feron et al 2013) in the part 2.3 of the materials and methods.

Figures : All the figures are too small and difficult to read. The quality of the immunofluorescence presented in figure 3 can be improved.

Authors should pay attention also because some typographic mistakes (extra spaces)...

Author Response

R2: The authors realized nasal biopsies to generate neurospheres (NS) from which they obtain neurospheres-derived cells (NDC). The authors characterised and compare NS and NDC using immunofluroescence and RNA-seq expression profile. The authors found that the NDC are in a transition state of differenciation from a glial/neural to a vascular pre-profile. They also discover that the NDC exert some metabolic features, that they compared to neuronal progenitor treated with a fongicide (maneb). The authors concluded that additional protocols are needed to use NDC for mass sample collection in psychatric diseases.

The research was conducted correctly and the authors present possible further steps that are necessary to accomplish to verify their findings. They also point out the limitations of their study.

I have minor suggestions:

Introduction - The authors should provide a mors detailed description and/or explanation of the cells they considers. Indeed, it is really confusing to cite olfactory ensheating cells (OEC) without mentionnig the olfactory stem cells (OSC). The end of the introduction has to be revised according to this suggestion. The confusion is reinforced later because authors described olfactory stem cells obtention (Feron et al 2013) in the part 2.3 of the materials and methods.

A: Thank you for pointing this out. We mistakenly stated that our goal is to get OEC. In part the confusion comes from the literature where the term “OEC” is used interchangeably for native olfactory glia-like cells as well for cultivated cells from biopsies just like our NDC. We would be happy to get pure glial-like cell cultures as well as neuronal progenitors, but unfortunately NDC are in general not either of them. We made this point explicitly in the Introduction (“The aim of this work is to characterize these cells and, if possible, understand how the protocol can be modified to yield more cells with neuronal or glial identity”) and Discussion of the revised manuscript, and removed the notion of OEC from the Abstract.

R2: Figures : All the figures are too small and difficult to read. The quality of the immunofluorescence presented in figure 3 can be improved.

A: We redrew the pictures to improve resolution and readability.

R2: Authors should pay attention also because some typographic mistakes (extra spaces)...

A: We tried our best to get rid of typography mistakes in the revised version of the manuscript.

Reviewer 3 Report

In the manuscript entitled “Characterisation of Neurosphere-Derived Cells from Human  Olfactory Epithelium” Zelenova and colleagues aimed to characterize neurospheres (NS) and neurospere-derived cells (NDC) cells obtained from the olfactory epithelium of healthy individuals. This is an in interesting study, however I have reservations regarding the general idea of using NS and NDC as biomarkers for psychiatric disorders, what has been suggested by the Authors. See all my comments below.

Major comments

L85: Authors wrote: “The purpose of this work is to characterise NDC and estimate their utility in psychiatric research.” How the Authors can suggest that the studied NDC could be useful in psychiatric research if they did not collect them from mentally ill patients/compared to healthy controls? I am not questioning the need for identification of biomarkers useful for psychiatric research, however, the authors cannot draw this conclusion based on the results of this study. Only the first part of the goal has been achieved in the present study.

If the modification of cell growth conditions impacts differentiation of NS and can “redirect” them towards different cell populations (additionally, taking into account, what the Authors have also mentioned, that there are differences between laboratories in terms of protocols for cell culturing, etc.), it influences gene expression and signalization pathways – how this can reflect in vivo conditions or the disease state in patients with psychiatric illnesses? If other application of these cells was considered – it should be clearly stated/discussed.

Minor comments

L183-184 in legend to Fig.1 authors wrote “In all the panels except the first two the blue colour corresponds to DAPI dye” – I can see positively marked “blue” cells in the second panel. Aren’t these cells positive for DAPI? Pls verify. If not DAPI – what marker is it?

In Fig.3B, Fig.S1 (left bottom part), and Fig.S2 -  some of inscriptions are overlapping and, thus, are illegible. I appreciate complexity of the figures (it is not easy to put there such large amount of data), but the inscriptions should be readable.

Author Response

R3: In the manuscript entitled “Characterisation of Neurosphere-Derived Cells from Human  Olfactory Epithelium” Zelenova and colleagues aimed to characterize neurospheres (NS) and neurospere-derived cells (NDC) cells obtained from the olfactory epithelium of healthy individuals. This is an in interesting study, however I have reservations regarding the general idea of using NS and NDC as biomarkers for psychiatric disorders, what has been suggested by the Authors. See all my comments below.

Major comments

L85: Authors wrote: “The purpose of this work is to characterise NDC and estimate their utility in psychiatric research.” How the Authors can suggest that the studied NDC could be useful in psychiatric research if they did not collect them from mentally ill patients/compared to healthy controls? I am not questioning the need for identification of biomarkers useful for psychiatric research, however, the authors cannot draw this conclusion based on the results of this study. Only the first part of the goal has been achieved in the present study.

A: The point of the work was to find out if it is possible to get easily available neuronal cells from olfactory epithelium. If there are no neuronal cells to collect, it is hard to justify the hassle — blood cells are not neuronal and they are much more easily available. But we understand the concern and reformulated the purpose paragraph in the Introduction:

“The existing protocols involve obtaining neuronal or glial stem cells from neuro-spheres (NS) that are to be grown from nasal biopsy material on a neuronal selective medium. In a standard medium with serum, cells arise from NS by enzymatic dissociation using trypsin to propagate into an abundant monolayer of cells within two to three weeks, which we further refer to as neurosphere-derived cells (NDC). The aim of this work is to characterize these cells and, if possible, understand how the protocol can be modified to yield more cells with neuronal or glial identity.”

R3: If the modification of cell growth conditions impacts differentiation of NS and can “redirect” them towards different cell populations (additionally, taking into account, what the Authors have also mentioned, that there are differences between laboratories in terms of protocols for cell culturing, etc.), it influences gene expression and signalization pathways – how this can reflect in vivo conditions or the disease state in patients with psychiatric illnesses? If other application of these cells was considered – it should be clearly stated/discussed.

A: The Reviewer raises an interesting point. In functional psychiatric research, there is a tradeoff between availability, relevance and a native state of a cell material. Unfortunately, the most relevant and native cells are inside the foetal brain of future psychiatric patients, which are very hard to procure. In case of cells from nasal biopsies it is possible to trade relevance (whatever you do, you still have more relevant cells than blood) for native state (whatever you do, you still have more native cells than IPSC-neurons) while retain good availability (whatever you do, you still have more available cell material than post mortem brains).

We do not plan to use NDC for any applications other than psychiatric research. They could probably be considered in a context of regenerative medicine but that is not our intention to explore this. 

R3: Minor comments

L183-184 in legend to Fig.1 authors wrote “In all the panels except the first two the blue colour corresponds to DAPI dye” – I can see positively marked “blue” cells in the second panel. Aren’t these cells positive for DAPI? Pls verify. If not DAPI – what marker is it?

A: We meant the first two panels of the top row, we rewritten the capture of Figure 1 to make it clear: “On all panels where the blue colour is present, it corresponds to DAPI dye”.

R3: In Fig.3B, Fig.S1 (left bottom part), and Fig.S2 -  some of inscriptions are overlapping and, thus, are illegible. I appreciate complexity of the figures (it is not easy to put there such large amount of data), but the inscriptions should be readable.

A: We updated the figures to make them more readable.

This manuscript is a resubmission of an earlier submission. The following is a list of the peer review reports and author responses from that submission.